

# Regulation of lipid metabolism by APOE4 in intrahepatic cholangiocarcinoma *via* the enhancement of ABCA1 membrane expression

Liqiang Qian[1,2], Gang Wang[1], Bin Li[1], Haoyuan Su[1] and Lei Qin[2]

[1] Department of General Surgery, Suzhou Ninth Hospital Affiliated to Soochow University, Suzhou, Jiangsu, China

[2] Department of General Surgery, The First Affiliated Hospital of Soochow University, Suzhou, Jiangsu, China

## ABSTRACT

Intrahepatic cholangiocarcinoma (ICC) is a malignancy with a dismal prognosis, thus the discovery of promising diagnostic markers and treatment targets is still required. In this study, 1,852 differentially expressed genes (DEGs) were identified in the GSE45001 dataset for weighted gene co-expression network analysis (WGCNA), and the turquoise module was confirmed as the key module. Next, the subnetworks of the 1,009 genes in the turquoise module analyzed by MCODE, MCC, and BottleNeck algorithms identified nine overlapping genes (CAT, APOA1, APOC2, HSD17B4, EHHADH, APOA2, APOE4, ACOX1, AGXT), significantly associated with lipid metabolism pathways, such as peroxisome and cholesterol metabolism. Among them, APOE4 exhibited a potential tumor-suppressive role in ICC and high diagnostic value for ICC in both GSE45001 and GSE32879 datasets. *In vitro* experiments demonstrated Apolipoprotein E4 (APOE4) overexpression suppressed ICC cell proliferation, migration, and invasion, knockdown was the opposite trend. And in ICC modulated lipid metabolism, notably decreasing levels of TG, LDL-C, and HDL-C, while concurrently increasing the expressions of TC. Further, APOE4 also downregulated lipid metabolism-related genes, suggesting a key regulatory role in maintaining cellular homeostasis, and regulating the expression of the membrane protein ATP-binding cassette transporter A1 (ABCA1). These findings highlighted the coordinated regulation of lipid metabolism by APOE4 and ABCA1 in ICC progression, providing new insights into ICC mechanisms and potential therapeutic strategies.

## BACKGROUND

Intrahepatic cholangiocarcinoma (ICC) is an exceptionally aggressive malignancy originating in the liver's intrahepatic biliary system (*Conci et al., 2018*). It is mainly associated with biliary tract diseases, for example, sclerosing cholangitis and hepatolithiasis (*Rizvi et al., 2018*). Despite accounting for merely 10% to 20% of primary liver cancers, the escalating incidence of ICC worldwide is still a cause for concern. The

Corresponding author
Lei Qin, qinleifyy163@163.com

primary etiological factors associated with ICC predominantly revolve around chronic liver diseases, like cirrhosis, and biliary tract disorders like primary sclerosing cholangitis, although a substantial fraction of cases remain idiopathic (*Karlsen et al., 2017*). Despite ongoing research efforts aiming to elucidate the multifaceted molecular and cellular mechanisms underlying ICC, its pathogenesis remains intricately complex (*Liu et al., 2022*). Presently, the most promising curative treatment approach is surgical resection, however, its efficacy is often compromised by the advanced stage of the disease at diagnosis (*Buettner et al., 2017*). Regrettably, the five-year survival rate for patients with ICC presently stands at less than 15% (*Rosen, Heimbach & Gores, 2010*), There was an urgent need for advancements in early detection and the enhancement of therapeutic strategies for hepatocellular carcinoma. Tumor biomarkers are pivotal in the early screening, diagnosis, treatment evaluation, recurrence monitoring, and prognosis prediction of these tumors. The early detection of hepatocellular carcinoma significantly impacts patient prognosis and treatment outcomes. Thus, the sensitivity and specificity of these biomarkers are crucial for the timely and accurate diagnosis of this malignancy (*Gao et al., 2020*). In both domestic and international clinical settings, alpha-fetoprotein (AFP), AFP-L3 isoform, and des-gamma-carboxy prothrombin (DCP) are recognized as prominent serum biomarkers for the early detection of hepatocellular carcinoma (*Yamashita et al., 2008*). Carbohydrate antigen 19-9 (CA19-9), initially discovered in the culture medium of a human colorectal cancer cell line, is extensively employed as a diagnostic marker for various adenocarcinomas, including cholangiocarcinoma. It serves as a crucial serological biomarker in ICC (*Qin et al., 2004*). Gene-targeted therapies have increasingly demonstrated promising therapeutic outcomes, as evidenced by numerous studies. Importantly, insights from integrative genomic research have pinpointed isocitrate dehydrogenase (IDH) as a potential therapeutic target for ICC. This enzyme facilitated the transformation of isocitrate to $\alpha$-ketoglutarate, with IDH1 operating in the cytoplasm and IDH2 in the mitochondria. Remarkably, in a specific study, mutations in IDH1 and IDH2 were identified in 10% of ICC patients. Furthermore, these mutations were linked with an extended period before tumor recurrence post-ICC resection. Consequently, such genetic biomarkers in hepatocellular carcinoma could significantly guide personalized gene-targeted treatments, enhancing the prognosis for patients (*Wang et al., 2013*). In the future, the identification of novel tumor biomarkers holds promise for advancing the early detection and targeted treatment of liver cancer. A deeper exploration of these biomarkers can elucidate the mechanisms underlying the onset and progression of ICC, paving the way for timely diagnosis and consequently enhancing the quality of life and survival rates of ICC patients.

The complex interplay between lipid metabolism and cancer pathogenesis, including ICC, has gained increasing attention in recent years (*Zhang et al., 2020*). Lipid metabolism, comprising lipid synthesis, oxidation, and trafficking, is essential in maintaining cellular homeostasis (*Ademowo et al., 2017*). Aberrations in lipid metabolic pathways can provide cancer cells with necessary substrates for biomass accumulation, energy production, and signaling molecule synthesis, thereby driving tumorigenesis and tumor progression (*Finley & Haigis, 2012*; *Alannan et al., 2020*). It had been shown that ATP-binding cassette transporter protein A1 (ABCA1) was a key protein in the regulation of cholesterol efflux

and was essential for high-density lipoprotein (HDL) formation. In amyloid precursor protein (APP) transgenic mice, ABCA1 deficiency resulted in increased amyloid deposition in the brain, along with reduced apolipoprotein E (APOE) levels. ABCA1 deficiency was found to significantly reduce Aβ clearance in mice expressing APOE4 by *in vivo* assays. The findings suggest that the presence of functional ABCA1 significantly affects the phenotype of APP transgenic mice expressing human APOE4 (*Fitz et al., 2012*). And findings suggested that the pathologic role of apoE4 *in vivo* may be related to reduced ABCA1 activity and impaired APOE4 lipids. Downstream brain-related pathology and cognitive deficits may be counteracted by treatment with the ABCA1 agonist CS-6253. These findings have important clinical implications and present ABCA1 as a promising target for ApoE4-related treatment of AD2 (*Boehm-Cagan et al., 2016*). The background of ICC makes the study of abnormal lipid metabolism particularly compelling due to the central role of the liver in systemic lipid metabolism (*Li et al., 2021*). ICC cells have been found to exhibit significant alterations in lipid metabolic profiles, such as elevated levels of triglycerides and cholesterol, which are potential contributors to the disease's severity and progression (*Paul, Lewinska & Andersen, 2022*). Furthermore, lipotoxicity, a condition emanating from excessive fatty acid accumulation, can inflict cellular damage and stimulate inflammatory responses, thereby contributing to ICC pathogenesis (*Anstee et al., 2019*; *Yamaguchi et al., 2022*). Thus, understanding the intricate mechanisms underpinning the interconnections between lipid metabolism and ICC pathogenesis not only offers valuable insights into disease biology but also opens up new avenues for therapeutic strategies targeting lipid metabolic pathways.

The purpose of this research was to elucidate the multifaceted role of lipid metabolism, specifically the role of APOE4 and ABCA1, in the pathogenesis of ICC. This study leverages high-throughput data analysis and experimental validation to shed light on the interaction between these key regulatory genes and their implications on lipid metabolic alterations in ICC. APOE4 down-regulates the levels of lipid metabolism-related genes TG, TC, LDL-C and HDL-C, and plays a key regulatory role in maintaining cellular homeostasis and regulating the expression of ABCA1. The insights derived from this investigation have substantial implications for disease diagnosis, prognosis, and therapeutic interventions. In essence, the study significantly contributes to our understanding of lipid metabolism dysregulation in ICC, paving the way for innovative therapeutic strategies targeting lipid metabolic pathways.

## MATERIALS AND METHODS

### Acquisition and analysis of datasets

The datasets GSE45001 (10 tumor and 10 normal samples) and GSE32879 (30 tumor and seven normal samples) were procured in the gene expression omnibus (GEO) database. Bioinformatics analysis was performed by the "limma" package in R software for the detection of differentially expressed genes (DEGs) from the GSE45001 dataset. A stringent selection criterion was set to identify DEGs with a $\log_2$ fold change ($\log_2$FC) greater than 1 or less than $-1$, and *P*-value less than 0.01, ensuring the reliability of the DEGs. The

statistical outputs were then processed, and data visualization was performed using the "ggplot2" package in R software.

## Weighted gene co-expression network analysis of DEGs

In the next step of the methodology, the DEGs derived from the GSE45001 dataset were subjected to weighted gene co-expression network analysis (WGCNA) for further understanding of the gene co-expression networks. To guarantee a scale-free network, an optimal soft-thresholding power ($\beta$) was determined through analysis of network topology. With the soft-thresholding power, to measure the network connectivity of a gene, which was defined as the sum of its adjacency with all other genes for network formation, an adjacency matrix was first produced and then translated into a topological overlap matrix (TOM). Genes were further separated into various modules after we performed average linkage hierarchical clustering based on TOM dissimilarity. Following the establishment of the gene co-expression network, the relation between the gene modules and the clinical traits of the tumor and normal samples within the GSE45001 dataset was calculated. The module with the strongest correlation coefficient was selected for subsequent analyses.

## Protein–protein interaction network construction

To probe the potential interactions among the 1,009 genes identified within the turquoise module, we implemented the Cytoscape software to construct a protein-protein interaction (PPI) network. Subsequently, we employed the molecular complex detection (MCODE), maximal clique centrality (MCC), and BottleNeck algorithms available within the Cytohubba plugin for distinct visualizations of the network, thereby facilitating the extraction of critical network modules. This analysis allowed us to visualize and interpret the potential interactions and pivotal nodes within the network. Subsequently, overlapping genes within the constructed networks were determined through the "VennDiagram" package in R software. The overlapping genes hold significance as they may represent critical junctures where multiple pathways converge. To gain a deeper understanding of the functional roles of these overlapping genes, we utilized the "ClusterProfiler" package for the Kyoto Encyclopedia of Genes and Genomes (KEGG) pathway analysis. Statistical significance was determined by a $P$-value below 0.05, reinforcing the validity of the obtained results.

## Differential expression analysis of overlapping genes and receiver operating characteristic curve evaluation

After the network construction, we delved into the expression analysis of nine overlapping genes within both tumor and normal samples in the GSE45001 and GSE32879 datasets. The statistical evaluation of these results underscored findings with a $P$-value below 0.05, indicative of a significant discrepancy in gene expression. Thereafter, we performed receiver operating characteristic (ROC) curve analysis for those genes demonstrating significant expression differences within the GSE32879 and GSE45001 datasets. This analysis served as a critical tool in appraising the diagnostic value of these genes concerning intrahepatic cholangiocarcinoma. As an evaluative measure of each gene's potential as a diagnostic biomarker, we calculated the area under curve (AUC). An AUC value nearing 1.0 is

indicative of an optimal diagnostic biomarker, whereas an AUC value of 0.5 signifies a lack of discriminatory power. Through the amalgamation of these analyses, we intended to pinpoint the key candidate gene that would serve as a significant factor in ICC diagnostics.

## Cell culture

The human ICC cell lines (RBE, CCLP1, HuCCT1, QBC-939, HCCC-9810) and the normal cholangiocarcinoma cell line (H69) were acquired from Cell Bank of the Chinese Academy of Sciences. ICC cell lines were kept in DMEM supplemented with 10% FBS, 100 U/mL penicillin, and 100 $\mu$g/mL streptomycin at 37 °C containing 5% $CO_2$. The H69 cell line was grown in RPMI 1640 medium with 10% FBS and the same antibiotics.

## Cell transfection

To construct an overexpression model of APOE4 in ICC cell lines, we implemented a plasmid transfection approach. Specifically, the APOE4 plasmid, along with an appropriate control plasmid, was introduced into the ICC cell lines utilizing Lipofectamine 2000 (Invitrogen, Carlsbad, CA, USA) based on the provided instructions. Following transfection, cells were allowed to grow for 48 h. To explored the effect of APOE4 downregulation, we used three different small interfering RNAs (siRNAs) targeting APOE4 for gene knockdown. These siRNAs were named si-APOE4-1, si-APOE4-2, and si-APOE4-3, which induced their degradation and reduced protein levels. ICC cells were transfected with si-APOE4 variants using Lipofectamine 2000 (Invitrogen, Carlsbad, CA, USA). To maintain the consistency of the experiment and to verify the knockdown efficiency, non-targeting siRNA was added as a negative control. After transfection, cells were incubated for an additional 72 h to ensure adequate knockdown, and then harvested for subsequent analysis. In parallel, to engineer a knockdown model of ABCA1 in ICC cell lines, cells were transfected with ABCA1-specific small interfering RNA (siRNA) again utilizing Lipofectamine 2000 (Invitrogen, Carlsbad, CA, USA). Non-targeting siRNA was applied as a control to ensure the specificity of the knockdown effects. After 72 h of post-transfection incubation, cells were collected for further analysis.

## Quantitative real-time polymerase chain reaction (qRT-PCR)

Total RNA was isolated from cells according to the manufacturer's instructions using a TRIzol reagent (Invitrogen, Carlsbad, CA, USA). Reverse transcription was conducted using the PrimeScript RT Reagent Kit (TaKaRa, Shiga, Japan). Subsequent quantitative real-time polymerase chain reaction (qRT-PCR) was performed by TB Green Premix Ex Taq II (Tli RNaseH Plus) on an Applied Biosystems 7500 Real-Time PCR System. The primer sequences used for amplification were referred to Table S1. GAPDH, the housekeeping gene, was used to normalize gene expression. The $2^{-\Delta\Delta CT}$ approach was used to calculate the relative expression of each gene.

## Western blotting

The cells were lysed in RIPA buffer containing protease and phosphatase inhibitors. The BCA Protein Assay Kit (Pierce, Waltham, MA, USA) was used to determine protein concentrations. SDS-PAGE was used to separate equal quantities of protein, which was

then transferred to PVDF membranes (Millipore, Burlington, MA, USA). Membranes were blocked with 5% nonfat milk and incubated with primary antibodies overnight at 4 °C, anti-APOE4 (Abcam, Cambridge, UK; catalog #ab279714,1:1000), anti-ABCA1 (Abcam, Cambridge, UK; catalog #ab18180,1:1000), anti-FAS (Abcam, Cambridge, UK; catalog #ab133619,1:1000), anti-ACC (Abcam, Cambridge, UK; catalog #ab109368,1:1000), anti-SCD1 (Abcam, Cambridge, UK; catalog #ab236868,1:1000), anti-GPT-1 (Abcam, Cambridge, UK; catalog #ab214179,1:1000), anti-SREBP1 (Abcam, Cambridge, UK; catalog #ab313881,1:500), and anti-PPARγ (Abcam, Cambridge, UK; catalog #ab310323,1:1000). Protein bands were detected using an enhanced chemiluminescence (ECL) kit (Thermo Scientific, Waltham, MA, USA) after incubation with suitable horseradish peroxidase-conjugated secondary antibodies.

### Cell counting kit-8 assay

The impact of APOE4 overexpression and knockdown on cell proliferation was assessed by the cell counting kit-8 (CCK-8) assay. After transfection, cells were put at standardized densities in 96-well plates and incubated at indicated time intervals (0 h, 24 h, 48 h, 72 h, and 96h). CCK-8 solution was introduced at each interval and the cells were further cultured for another 2 h. The optical density (OD) value, which correlates with the number of viable cells, was measured at 450 nm by a microplate reader.

### Transwell assays

Transfected cells were seeded into the upper chamber of the Transwell inserts, which were either coated (for invasion experiment) or uncoated (for migration assay) (*Zhang et al., 2021*). In the lower compartment, a chemo-attractant was injected, and cells were allowed to migrate or invade for a set amount of time. Cells that successfully migrated or infiltrated the membrane's bottom surface were labeled with DAPI and counted under a microscope to determine their number.

### Enzyme-linked immunosorbent assay

The impact of APOE4 overexpression on cellular lipid profile, as well as the effect of combined APOE4 overexpression and ABCA1 knockdown, were evaluated using enzyme-linked immunosorbent assay (ELISA). The levels of total cholesterol (TC), low-density lipoprotein cholesterol (LDL-C), triglycerides (TG), and high-density lipoprotein cholesterol (HDL-C) in the cell lysates were detected by specific ELISA kits.

### Statistical analysis

All results were represented as mean ± standard deviation (SD) from at least three independent experiments. Comparisons between two groups were studied using Student's *t*-test, while one-way analysis of variance (ANOVA) followed by Tukey's post-hoc test was utilized for multi-group comparisons. All statistical analyses were performed by GraphPad Prism software (Version 8.0, San Diego, CA, USA).

## RESULTS

### The turquoise module has the highest correlation with the GSE45001 data samples

By employing predefined $\log_2$FC and *P*-value thresholds, we identified 811 upregulated and 1,041 downregulated DEGs from the GSE45001 dataset (Fig. 1A). To decipher the complex relationships among these DEGs, WGCNA was performed. The network was constructed utilizing an optimal soft-thresholding power of 16, ensuring a scale-free network (Fig. 1B). DEGs were then categorized into different modules based on their expression patterns within the GSE45001 dataset (Fig. 1C). A cluster dendrogram was used for visualizing the network hierarchy, with distinct colors representing each module to facilitate differentiation (Fig. 1D). The adjacent eigengenes heatmap validated the independence of these modules, signifying their distinctive gene co-expression relationships (Fig. 1E). To further investigate the relationships between the gene modules and GSE45001 samples, module-trait correlation heatmaps were employed (Fig. 1F). Among all modules, the turquoise module showed the highest correlation with the samples in the dataset, with a correlation value of 0.935 (Fig. 1G). As such, it was the key module for subsequent in-depth analyses.

### Nine overlapping genes were identified by PPI networks on the turquoise module

With the help of Cytoscape software, a PPI network analysis was performed on the 1,009 genes present in the turquoise module. The algorithms of MCODE 1 (22 nodes and 231 edges), MCODE 2 (11 nodes and 51 edges), MCC (35 nodes and 291 edges), and BottleNeck (35 nodes and 291 edges) facilitated the visualization of key modules in the network. This resulted in a division into four main modules, as shown in Figs. 2A–2D. Subsequent screening within these networks yielded nine overlapping genes (Fig. 2E). To elucidate their potential functional roles, we set out to perform an enrichment analysis of these genes. Subsequent results revealed their important involvement in 10 KEGG pathways. These included peroxisome, cholesterol metabolism, Biosynthesis of unsaturated fatty acids, glyoxylate and dicarboxylate metabolism, beta-Alanine metabolism, propanoate metabolism, tryptophan metabolism, fatty acid degradation, PPAR signaling pathway, and primary bile acid biosynthesis (Fig. 2F).

### Significant downregulation of overlapping genes in ICC and the diagnostic value estimation

Subsequently, the expression of the overlapping genes was analyzed in samples from GSE45001 and GSE32879.As illustrated in Figs. 3A–3B, CAT, APOA1, APOC2, HSD17B4, EHHADH, APOA2, APOE4, ACOX1, and AGXT were found to be significantly reduced in the tumor samples of the GSE45001 dataset. However, in the GSE32879 dataset, only seven genes (APOA2, CAT, ACOX1, APOE4, AGXT, EHHADH, HSD17B4) were observed to be significantly reduced in tumor samples. Given their significant downregulation, we proceeded to assess the clinical diagnostic value of these seven genes across both the GSE32879 and GSE45001 datasets. As inferred from the ROC curves depicted in

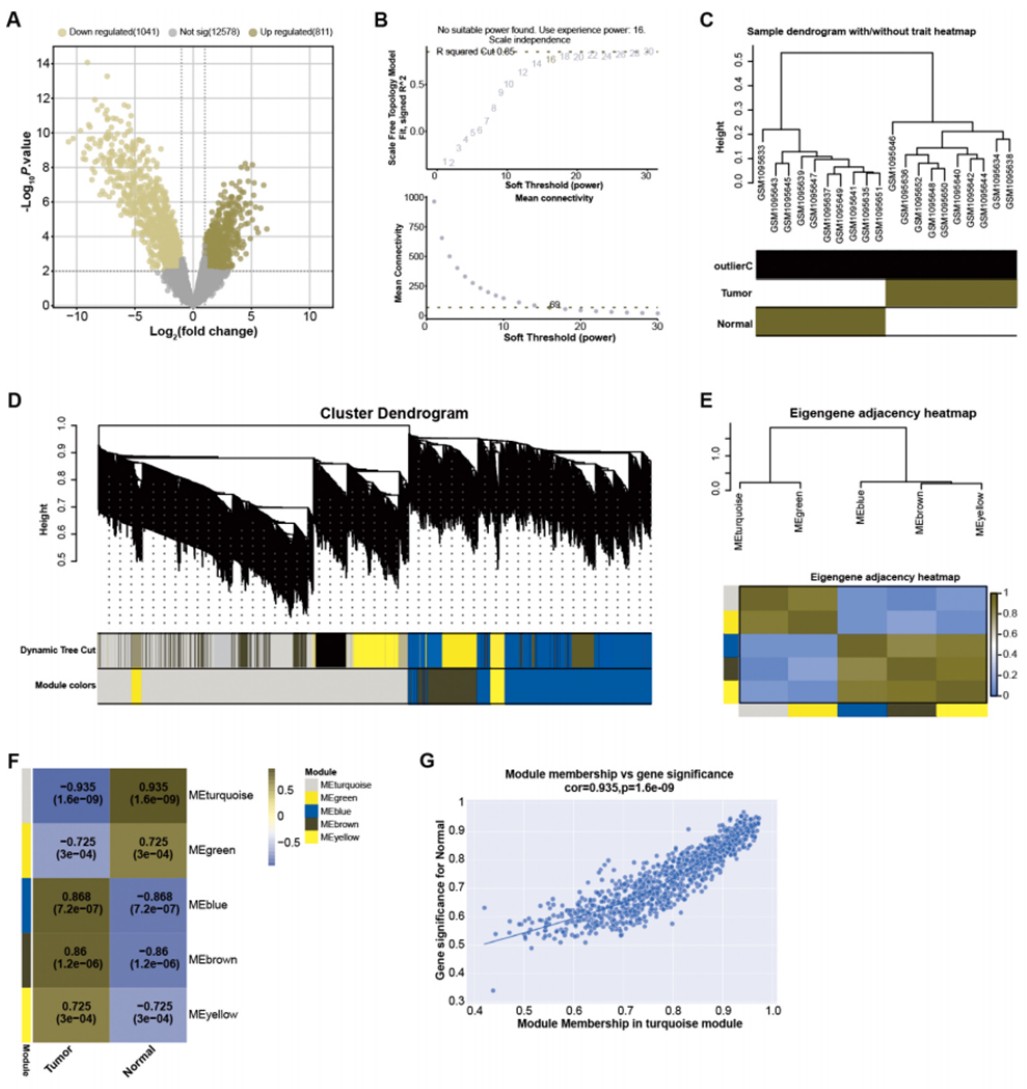

**Figure 1** Identification of DEGs and construction of gene co-expression network.

Figs. 3C–3D, the AUC for these genes surpassed the threshold of 0.7, thereby indicating their substantial diagnostic relevance in ICC. Among these significantly downregulated genes, APOE4 holds a crucial role in modulating multiple pathways, inclusive of lipid transport and metabolism, which are implicated in the onset of carcinogenesis. Owing to its potential relevance in disease pathogenesis, we resolved to investigate APOE4 in subsequent *in vitro* experiments to elucidate its potential role in the pathogenesis of ICC.

## APOE4 regulates the growth of ICC cells *in vitro*

Initially, we examined the level of APOE4 in ICC cell lines using qRT-PCR. Remarkably, the level of APOE4, at the mRNA level, was found to be considerably diminished in ICC cell lines when compared with their normal counterparts, with CCLP1 and HuCCT1 cell lines particularly standing out (Fig. 4A). This observation implicates APOE4 as a

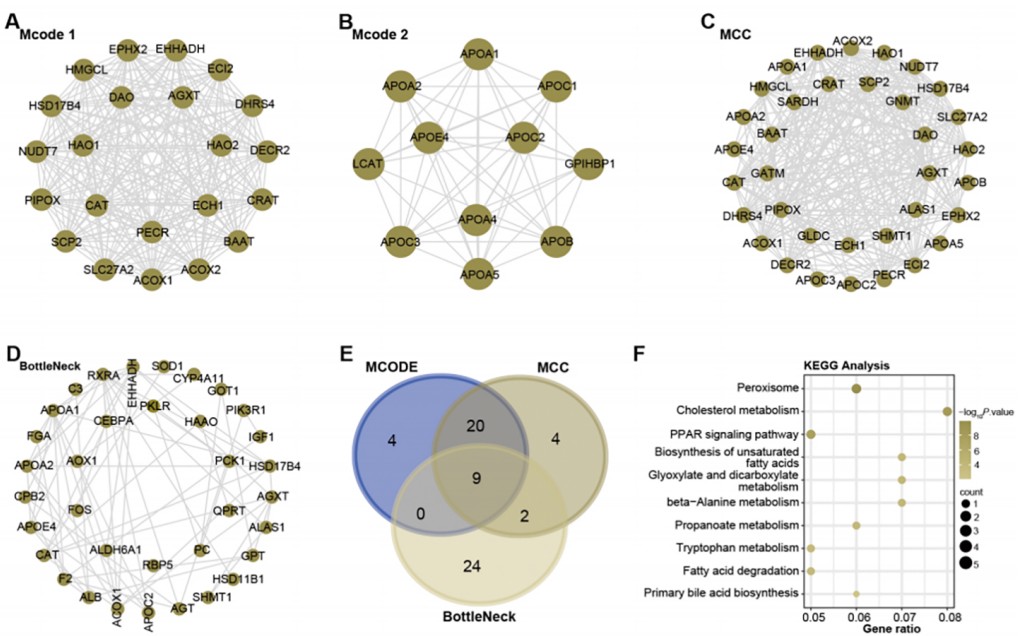

**Figure 2** Network and enrichment analysis of the turquoise module.

putative tumor suppressor in ICC. Further, we proceeded with the construction of APOE4 overexpression plasmids and knockdown of APOE4, subsequently transfected these into CCLP1 and HuCCT1 cell lines. Subsequent changes in APOE4 expression levels further confirmed our initial findings (Figs. 4B–4C, Figs. 1A–1B). Next, we performed CCK-8 and Transwell experiments to evaluate the effects of APOE4 overexpression and knockdown on function. As shown in Figs. 4D–4E, overexpression of APOE4 significantly inhibited the proliferation of ICC cells; while the results in Figs. 4F–4I indicated that overexpression of APOE4 significantly inhibited the migration and invasion of ICC cells. In contrast, knockdown of APOE4 significantly increased their proliferation, migration and invasion abilities (Figs. 1C–1H). This suggested that APOE4 may be a suppressor gene whose expression significantly affected the tumorigenic properties of ICC cells, providing useful insights for potential applications in ICC therapy.

## APOE4 regulates lipid metabolism and related genes in ICC cells

TG, LDL-C, TC, and HDL-C are well-known key indicators of lipid metabolism, and their aberrant levels are often detected in the pathogenesis of multiple diseases, including cancers (*Chi et al., 2014*). Consequently, their expression was investigated to unveil potential links between dysregulated lipid metabolism and ICC. Following overexpression of APOE4 in ICC cells, expressions of TG, LDL-C, and HDL-C decreased as detected by the ELISA assay. while an increase in the levels of TC (Figs. 5A–5H). This finding not only underscores the putative modulatory role of APOE4 in lipid metabolism but also suggested its potential involvement in the complex pathogenesis of ICC. Furthermore, we analyzed the regulation of lipid metabolism-related genes (FAS, ACC, SCD1, CPT11,

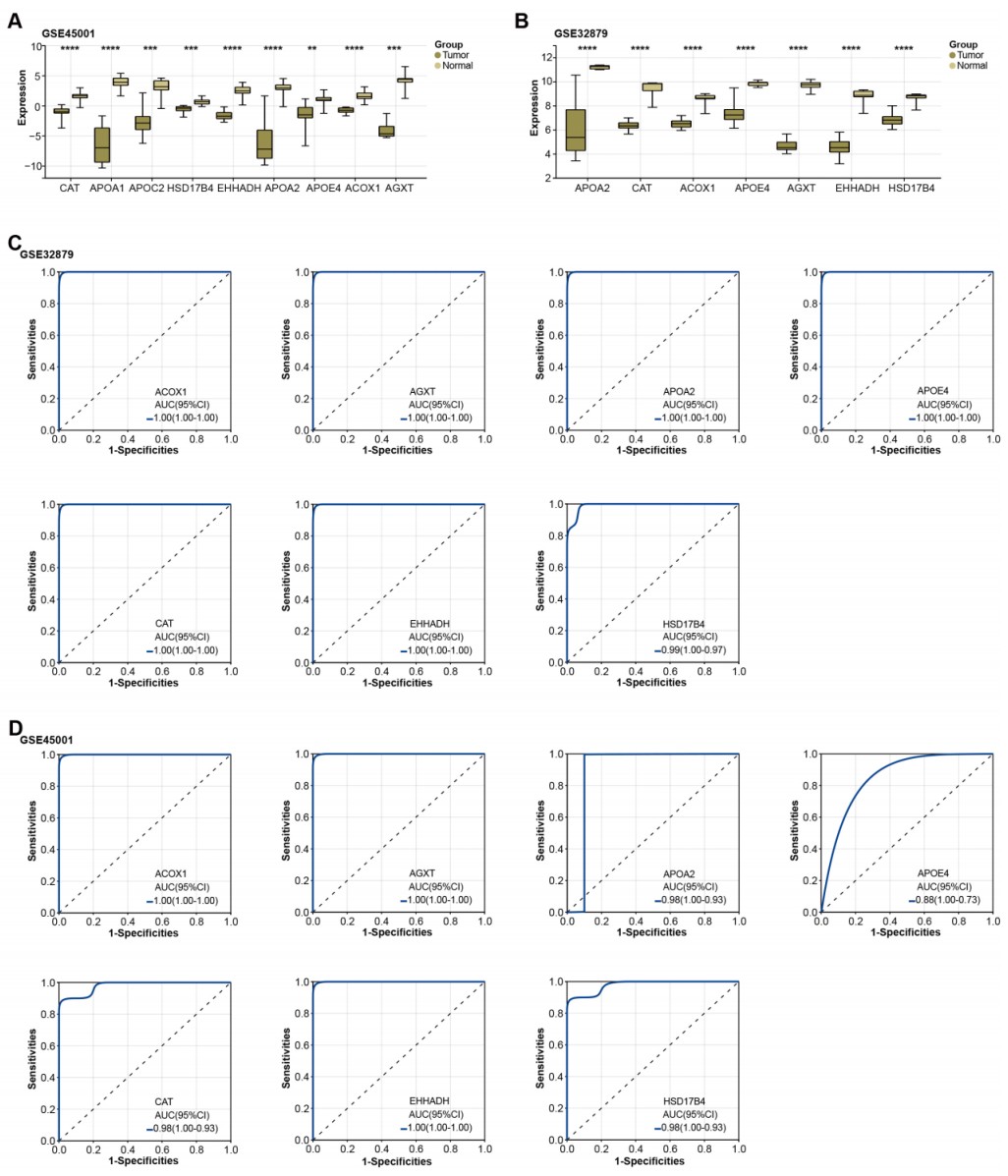

**Figure 3** Expression of overlapping genes and their clinical diagnostic value in ICC.

SREBR1, PPARγ) by APOE4. Analysis of RNA expression levels by qRT-PCR showed that overexpression of APOE4 resulted in a significant down-regulation of the expression of the relevant genes (Figs. 5I–5L). Western blotting analysis performed at the protein level yielded similar results, overexpression of APOE4 led to a significant reduction in the expression levels of these genes (Fig. 5M). Upon APOE4 knockdown, the trend was reversed, with increased expression of these genes (Fig. S1I). This implied that APOE4 could be a key regulatory gene in lipid metabolism, playing a pivotal role in maintaining cellular homeostasis and, in turn, preventing ICC development.

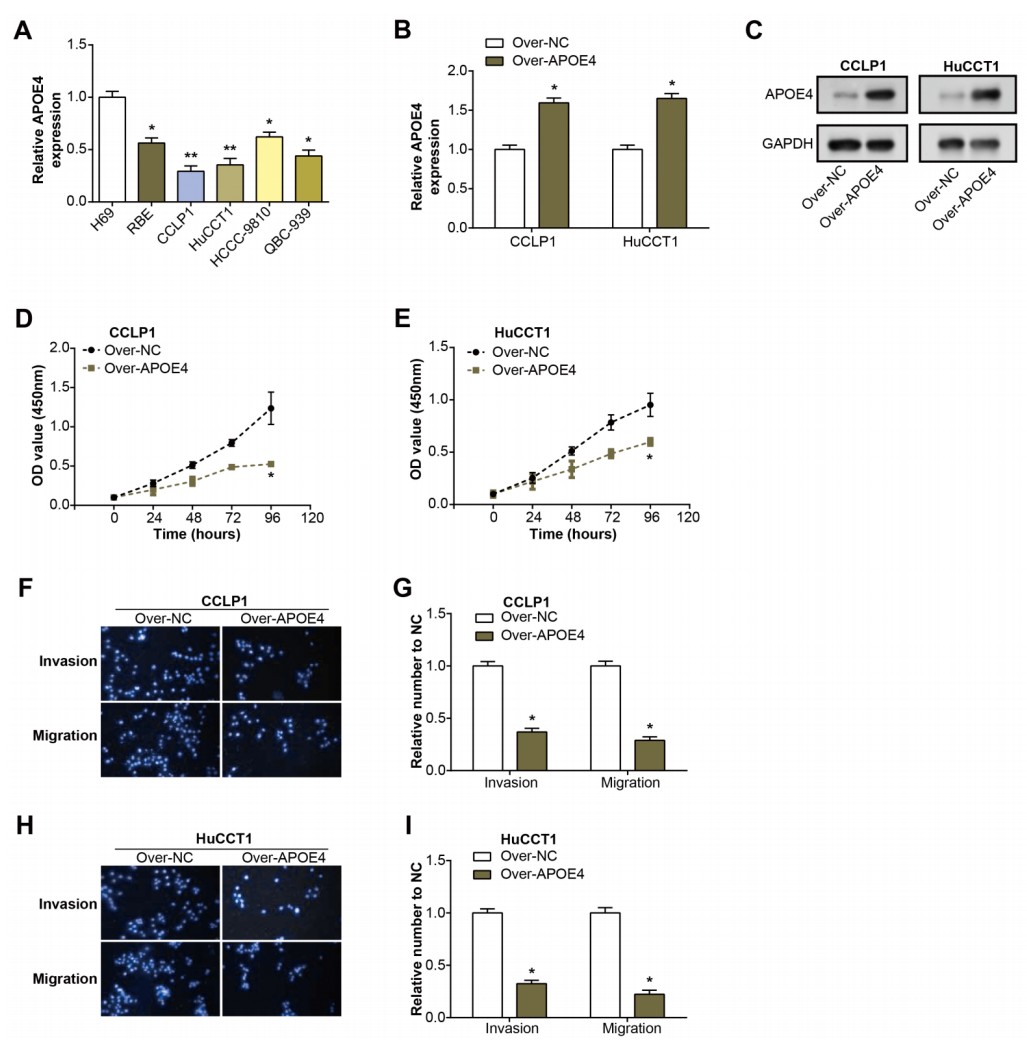

**Figure 4** **Expression and functional analysis of APOE4 in ICC cell lines.**

## APOE4 regulates lipid metabolism in ICC by enhancing the expression of the ABCA1 membrane

APOE4 has been found to modify ABCA1 membrane trafficking in astrocytes, enhancing cholesterol efflux by promoting its recycling to the plasma membrane (*Rawat et al., 2019*). ABCA1 primarily transports cholesterol and phospholipids to apolipoprotein receptor particles (*Zhou et al., 2015*). Hence, in our study, we specifically studied if ABCA1 was involved in the APOE4-mediated mechanism in ICC. After APOE4 overexpression in ICC cells, qRT-PCR analysis revealed upregulation of ABCA1 mRNA levels (Fig. 6A). Similarly, WB detected a significant increase in ABCA1 protein levels due to APOE4 overexpression (Fig. 6B). This suggested that APOE4 could play a crucial role in the regulation of cholesterol efflux, potentially by modulating the expression of ABCA1. Subsequently, the regulation of ABCA1 in cellular membrane and cytosolic components by overexpressed APOE4 in ICC cells was examined through WB. As depicted in Fig.

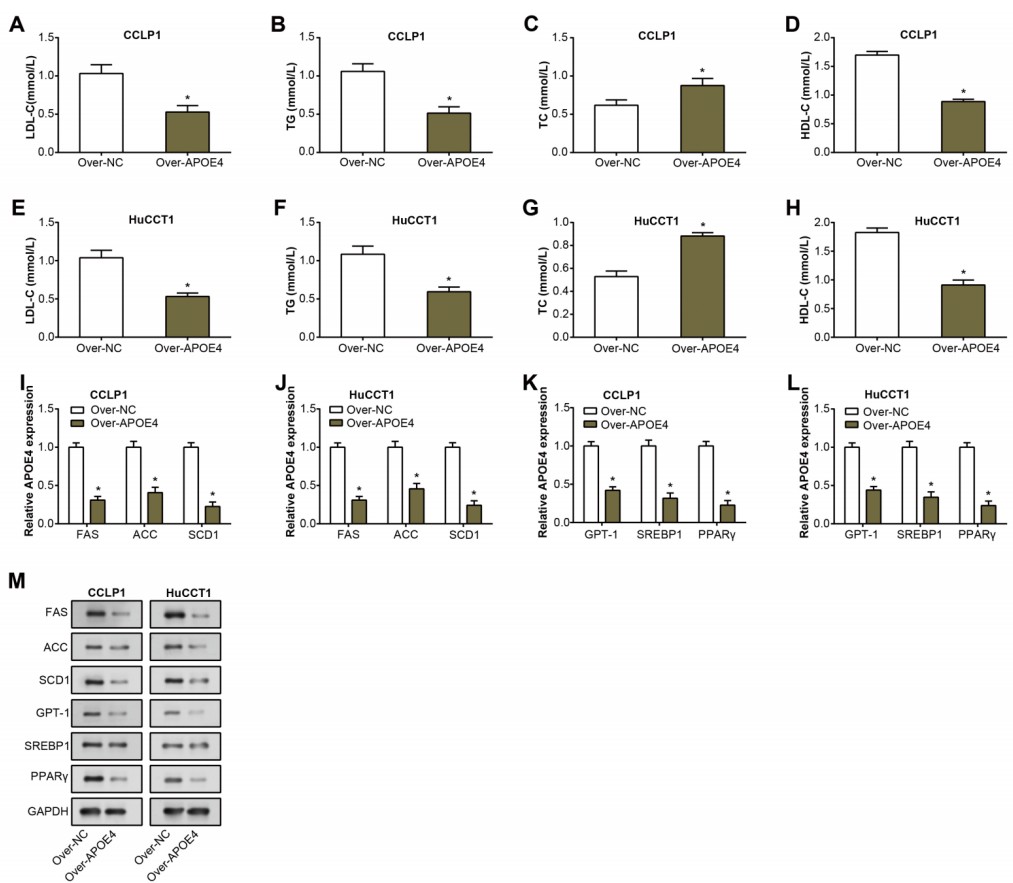

**Figure 5** APOE4 overexpression regulates lipid metabolism and downregulates lipid metabolism-related genes in ICC cells.

6C, there was an upregulation of ABCA1 expression in the cellular membrane, whereas no significant change was observed in the cytosolic components. This suggested that APOE4 may specifically influence the membrane trafficking of ABCA1, thus enhancing cholesterol efflux. Furthermore, ABCA1 was knocked down in ICC cells, leading to a considerable knockdown efficiency (Figs. 6D–6E). ELISA results confirmed a decrease in the concentrations of TG, LDL-C, and TC following APOE4 overexpression in ICC cells. Interestingly, we noticed a subsequent rise in these levels after ABCA1 knockdown (Figs. 6F–6L). In contrast, HDL-C levels manifested an elevation consequent to APOE4 overexpression in ICC cells. In addition, it was found that downregulation of ABCA1 could partially attenuate the effect of APOE4 overexpression on HDL-C concentration in ICC cells (Figs. 6I–6M). This indicates that APOE4 and ABCA1 collaboratively regulate lipid metabolism in ICC cells, and the suppression of ABCA1 could potentially counteract the lipid-lowering effect of APOE4 overexpression.

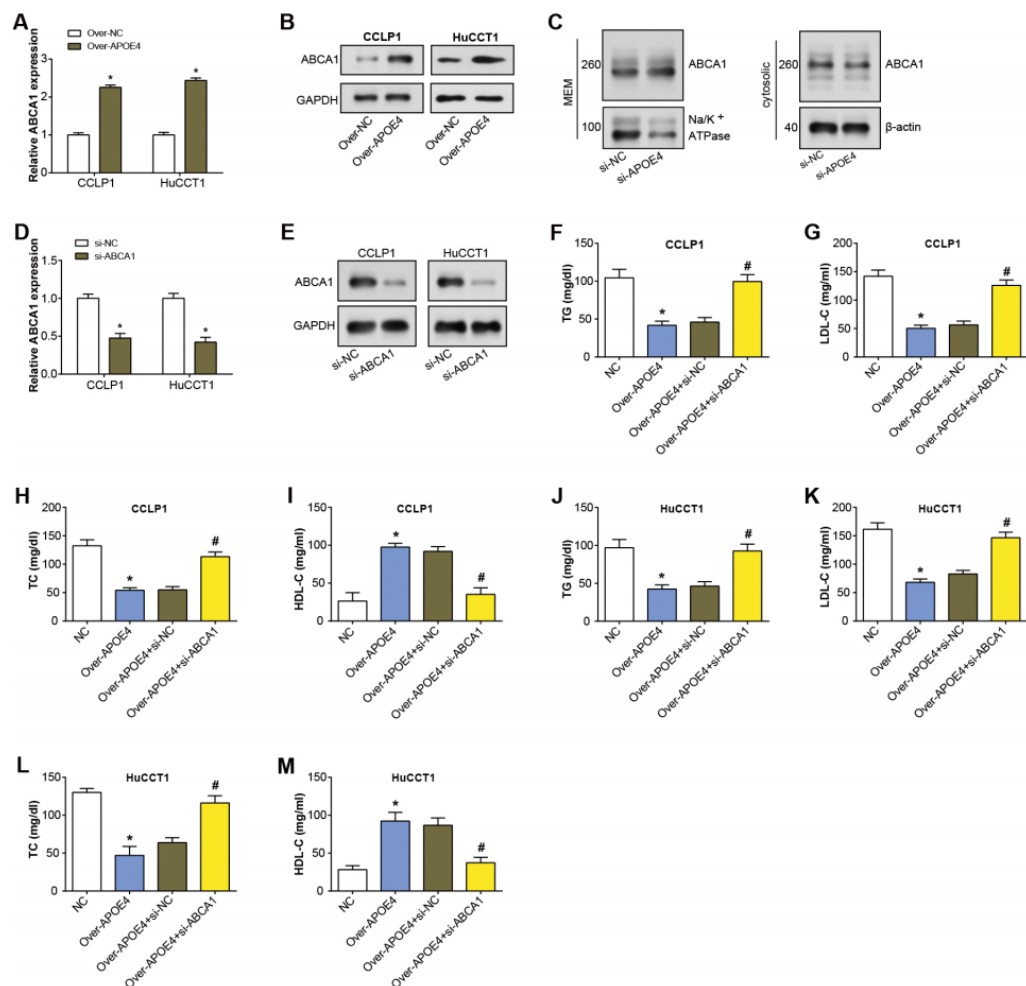

**Figure 6** APOE4 overexpression regulates ABCA1 expression and lipid metabolism in ICC cells.

## DISCUSSION

ICC remains one of the most challenging malignancies due to its aggressive nature and poor prognosis (*Chen et al., 2022*). In most cases, surgical resection is viewed as the only radical treatment option; however, many patients with ICC are often at an advanced stage at the time of their initial visit, and surgical resection is not a viable option (*Nakeeb et al., 1996*; *Shaib & El-Serag, 2004*). In addition, although surgery provides short-term therapeutic benefits for some patients, up to 70% of patients remain at risk of tumor recurrence after radical resection, and once recurrence occurs, their survival rate is significantly reduced (*Spolverato et al., 2016*). Therefore, over the past years, researchers have been investigating alternative nonsurgical treatments to improve the survival of those patients with ICC who are not suitable for surgery, their survival rate will be significantly reduced.5 Therefore, over the past years, researchers have been exploring alternative non-surgical therapies to improve the survival of ICC patients who are not surgical candidates.

However, despite a large number of studies, most studies on nonsurgical therapies have failed to demonstrate satisfactory results in the treatment of ICC. Among the studies on ablative therapies for solid tumors, radiofrequency ablation (RFA) is increasingly becoming an option for the treatment of liver tumors, and its efficacy in the treatment of hepatocellular carcinoma has been confirmed (*Shiina et al., 2012*; *Tiong & Maddern, 2011*). In addition, current research had primarily focused on the genomic landscape of ICC, identifying potential driver mutations and aberrations that contribute to its pathogenesis (*Rizvi & Gores, 2017*). In parallel, emerging studies are elucidating the role of metabolic dysregulation in ICC, including lipid metabolism (*Du et al., 2022*). Several investigations have implicated perturbed lipid metabolism, especially the elevation of triglycerides and cholesterol, in the promotion of tumor progression and bolstering of cancer cell survival within the context of ICC (*Paul, Lewinska & Andersen, 2022*). Additionally, abnormalities in lipid metabolism have been proposed as potential diagnostic markers for ICC (*Ma, Feng & Zhou, 2020*). Therefore, it is likely that future studies will pay more attention to the study of the application of biomarkers in order to find a more desirable therapeutic approach. However, the specific mechanisms by which altered lipid metabolism contributes to ICC development and progression are still unclear. This study thus aims to elucidate these mechanisms, particularly focusing on the roles of APOE4 and ABCA1 in regulating lipid metabolic processes in ICC.

In this research, WGCNA was employed to investigate the DEGs in the GSE45001 dataset. The turquoise module emerged as a focal point, demonstrating the highest correlation with the dataset. This analysis elucidated complex patterns of DEG expression and its association with ICC, thereby enriching the understanding of the molecular landscape of the disease. Further application of network analysis and pathway enrichment to the turquoise module offered a detailed PPI network among its constituent genes. The identification of distinct subsets within the PPI network using different network analysis algorithms, and the discovery of nine overlapping genes (CAT, APOA1, APOC2, HSD17B4, EHHADH, APOA2, APOE4, ACOX1, AGXT) within these subsets. The pathway enrichment analysis revealed intriguing functional implications of these overlapping genes. These genes exhibited substantial enrichment in critical KEGG pathways such as peroxisome activity, cholesterol metabolism, and PPAR signaling, and others. This underlined their possible crucial roles in lipid metabolism, a key aspect in the pathogenesis of ICC. For instance, the cholesterol metabolism pathway has been previously linked with the progression of ICC (*Yu et al., 2023*). High cholesterol levels have been associated with increased ICC risk, and drugs targeting cholesterol metabolism have shown promise in preclinical ICC models. The peroxisome proliferator-activated receptor (PPAR) signaling pathway has also been implicated in ICC (*Armstrong et al., 2014*; *Lu, Han & Wu, 2013*). PPARs regulate genes involved in lipid and glucose metabolism and inflammation (*Montaigne, Butruille & Staels, 2021*), and dysregulation of PPAR signaling has been observed in ICC patients (*Liu et al., 2023*), further strengthening the case for these pathways' relevance to ICC. These findings not only illuminated these genes as potential therapeutic targets or diagnostic markers but also underscored the necessity for in-depth exploration of the roles and mechanisms of these genes within these pathways.

In tumor samples taken from the GSE45001 and GSE32879 datasets, seven genes (APOA2, CAT, ACOX1, APOE4, AGXT, EHHADH, and HSD17B4) showed a substantial downregulation. Among them, APOE4 appeared as a generally downregulated gene in both datasets. An evaluation of the diagnostic ability of these downregulated genes, inclusive of APOE4, unveiled promising results. The ROC curves analysis demonstrated an AUC exceeding 0.7, underscoring their potent potential as diagnostic biomarkers for ICC. APOE4 is known for its capacity to modulate lipid transport and metabolism (*Wang et al., 2022*). Previous research by *Sienski et al. (2021)* highlighted that APOE4 could induce lipid defects in iPSC-derived human astrocytes. Other research pointed out that the cholesterol dysmetabolism prompted by APOE4 might mediate Alzheimer's disease-associated pathologies (*Huang et al., 2022*; *Jeong et al., 2019*). Additional literature has also underscored the role of APOE4 in carcinogenesis (*Hsu et al., 2019*). This intriguing connection led us to select APOE4 for further exploration in our *in vitro* cell experiments. Our principal aim was to illuminate its potential contributions to the pathogenesis of ICC, opening new avenues for diagnosing and managing this malignant tumor.

The findings elucidated the functional role of APOE4 in ICC pathogenesis and its potential as a therapeutic target. The suppressed expression of APOE4 in ICC cell lines indicated its tumor-suppressive role. Overexpression of APOE4 effectively curtailed ICC cell proliferation, migration, and invasion, supporting its role as a potential tumor suppressor. Further investigations into the influence of APOE4 on lipid metabolism unveiled a consequential effect. After APOE4 overexpression, decreased levels of LDL-C, TG, and HDL-C were observed, while TC levels increased. This modulation of the lipid profile underscored the potential role of APOE4 in the control of lipid metabolism in the context of ICC. In addition, the downregulation of lipid metabolism-related genes after APOE4 overexpression indicated that APOE4 was a key regulatory gene in lipid metabolism that helped maintain cellular homeostasis and thus hindered the development of ICC. In addition, the study illuminated the role of ABCA1, a pivotal component in lipid metabolism. It had been previously established that ABCA1 primarily facilitates the transport of cholesterol and phospholipids to apolipoprotein receptor particles, hence playing an instrumental role in maintaining lipid homeostasis (*Wang & Westerterp, 2020*; *Raulin, Martens & Bu, 2022*). Overexpression of APOE4 upregulated ABCA1 expression, indicating a potential role for APOE4 in the regulation of cholesterol efflux, possibly *via* modulating ABCA1 expression. Notably, ABCA1 knockdown counteracted the lipid-lowering effect of APOE4 overexpression, highlighting the collaborative regulation of lipid metabolism by APOE4 and ABCA1 in ICC cells. These findings provided a solid foundation for further exploration of the role of APOE4 and its interaction with ABCA1 in lipid metabolism and the pathogenesis of ICC.

## CONCLUSION

In conclusion, the study illustrated the crucial role of lipid metabolism and the APOE4 gene in the context of ICC. Dysregulated lipid metabolism emerges as a key facet in the pathogenesis of ICC, reinforcing the necessity to further decipher its complexities. In our

study, APOE4 was found to be significantly downregulated in ICC, and our investigations revealed its pivotal role as a potential tumor suppressor gene, particularly through its impact on cell proliferation, migration, invasion, and lipid metabolism. Specifically, APOE4 enhanced membrane ABCA1 expression to modulate lipid metabolism in ICC. This investigation provided valuable insights into the molecular mechanisms underpinning ICC and presented promising avenues for future research in ICC clinical strategies.

### Funding
The authors received no funding for this work.

### Competing Interests
The authors declare there are no competing interests.

### Author Contributions
- Liqiang Qian conceived and designed the experiments, prepared figures and/or tables, and approved the final draft.
- Gang Wang performed the experiments, authored or reviewed drafts of the article, and approved the final draft.
- Bin Li analyzed the data, prepared figures and/or tables, and approved the final draft.
- Haoyuan Su analyzed the data, authored or reviewed drafts of the article, and approved the final draft.
- Lei Qin analyzed the data, authored or reviewed drafts of the article, and approved the final draft.

### Data Availability
The raw data are available in the Supplemental Files.

### Supplemental Information
Supplemental information for this article can be found online at http://dx.doi.org/10.7717/peerj.16740#supplemental-information.

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
