# Peer review of "Regulation of lipid metabolism by APOE4 in intrahepatic cholangiocarcinoma via the enhancement of ABCA1 membrane expression"

_PeerJ, doi:10.7717/peerj.16740_

## Round 0.1 · original submission · Major Revisions

Please address the concerns of all reviewers and amend the manuscript accordingly.

Reviewer 1 has requested that you cite specific references. You may add them if you believe they are especially relevant. However, I do not expect you to include these citations, and if you do not include them, this will not influence my decision.

**Language Note:** The review process has identified that the English language must be improved. PeerJ can provide language editing services - please contact us at copyediting@peerj.com for pricing (be sure to provide your manuscript number and title). Alternatively, you should make your own arrangements to improve the language quality and provide details in your response letter. – PeerJ Staff

Reviewer 1 ·

Basic reporting

Dear Editor, thank you so much for inviting me to revise this manuscript.
A linguistic revision is needed before eventual publication. A professional service is recommended since it is not possible to accept the paper in its current linguistic form.

This paper addresses a current topic in CCA.

The manuscript is quite well written and organized. English should be improved.
Figures and tables are comprehensive and clear.
The introduction explains in a clear and coherent manner the background of this study.

We suggest the following modifications:
* Introduction section: although the authors correctly included important papers in this setting, we believe the evolving systemic treatment scenario for CCA (especially in terms of predictive biomarkers of response) should be briefly discussed and some recently published papers added within the introduction ( PMID: 36633661; PMID: 33592561; PMID: 35031442 ; PMID: 33756174 ), only for a matter of consistency. We think it might be useful to introduce the topic of this interesting study.

Experimental design

None to modify

Validity of the findings

none to modify

Additional comments

* Of note, the authors should expand some parts, including a more personal perspective to reflect on. For example, they could answer the following questions - in order to facilitate the understanding of this complex topic to readers: What are the knowledge gaps and how do researchers tackle them? How do you see this area unfolding in the next 5 years? We think it would be extremely interesting for the readers.

We believe this article is suitable for publication in the journal although major revisions are needed. The main strengths of this paper are that it addresses an interesting and very timely question and provides a clear answer, with some limitations.

We suggest a linguistic revision and the addition of some references for a matter of consistency. Moreover, the authors should better clarify some points.

Reviewer 2 ·

Basic reporting

In this study, the authors mainly focus on discovering promising diagnostic makers in intrahepatic Cholangiocarcinoma (ICC), which is a malignancy with a poor prognosis. Dependen on weighted gene co-expression network analysis (WGCNA) and turquoise module, Qian et al identified nine promising overlapping genes, which associated with the lipid metabolism. Specifically, APOE4 has a strong and significant tumor suppressive role in ICC. Mechanistically, APOE4 down regulates lipid metabolism-related genes and increases the expression of the membrane protein ABCA1. Therefore, the authors demonstrate APOE4 can inhibit tumor growth via lipid metabolism. Overall, the logic of this manuscript is clear and the data are convincing. However, there are some issues that should be addressed, thus the manuscript could not be accepted for publication in PeerJ in its current form.

Experimental design

APOE4 is down-regulated cancer as the authors demonstrated, which show APOEB as a potential tumor suppressive factor in ICC. And knock down APOE4 can obliviously inhibited the tuomrigenesis. So, how APOE4 was fereuqntly inhibited in ICC, and the authors should be analyzed using the same database if possible and discussed about these in this manuscript.

In Figure 5M, the author should test lipd metabolism-related genes change after knock down APOE4 in CCLP1 and HUCCT1 cell lines.

Validity of the findings

The quality of Figure 1 and 2 are very bad, the authors should improve them.

In Figure 6 F-M, the authors should make compare in a professional way. The original compare method is not good.

Additional comments

There are some spelling/grammatical errors in this manuscript, which should be corrected and modified. Such as line 160 “hours” and 185 “h”, should have a same format.

Reviewer 3 ·

Basic reporting

• The manuscript submitted by L Qian et al. studies the regulation of lipid metabolism by APOE4 and ABCA1 in ICC (Intrahepatic Cholangiocarcinoma) progression. Three major findings, first in this study using different algorithms, authors identified nine overlapping genes (CAT, APOA1, APOC2, HSD17B4, EHHADH, APOA2, APOE4, ACOX1, AGXT), significantly associated with lipid metabolism pathways in ICC progression. Second, using in vitro experiments, authors showed APOE4 overexpression suppressed ICC cell proliferation, migration, and invasion. Third, the authors showed APOE4 also downregulated lipid metabolism-related genes, suggesting a key regulatory role in maintaining cellular homeostasis and regulating the expression of the membrane protein ABCA1. The literature is cited well in the manuscript and written in the correct English.

Experimental design

• The abstract, discussion, and conclusion are clearly written with well-designed experiments and analysis. However, the manuscript has several major and minor issues that should be addressed before acceptance.
• It would be helpful to the reader if the authors included more details in the results section. The specific details are mentioned in the additional comments section.
• In Figures 1 and 2, please increase the font size to make it easily visible to the readers.

Validity of the findings

no comments

Additional comments

• It will be helpful for the reader if in the introduction section the authors incorporate what is already known in the literature about the correlation between APOE4 and ABCA1 in ICC progression (such as in reference 16). Please shed light on the novelty of the work.
• Line 90-92: In this statement, I would like the authors to specify the lipid metabolic pathways they want to point out for the therapeutic strategy such as mentioned in reference 21. I am also curious to know if any therapeutic approach is known that targets ABCA1 in the literature for any type of cancer.
• Line 177-178: Please catalog the antibodies and their dilutions used for the experiments, that will be helpful for the reproducibility of the experiments.
• Line 263-264: It would be helpful for the readers if the authors elaborate on Figure 4D-4I in detail.
• Line 278-280: Same as the previous statement, please elaborate on figure 5I-5M in detail.

---

## Round 0.2 · Minor Revisions

Please address the remaining concerns of the reviewers and amend the manuscript accordingly.

Reviewer 2 ·

Basic reporting

The authors have carefully revised the manuscript as the reviewers suggested.

Experimental design

Experimental design meet the journal's standards.

Validity of the findings

The findings are convincing.

Additional comments

After the revision, this manuscript can be published in this journal.

Reviewer 3 ·

Basic reporting

The manuscript submitted by L Qian et al. studies the regulation of lipid metabolism by APOE4 and ABCA1 in ICC (Intrahepatic Cholangiocarcinoma) progression.
After revision, the manuscript reads better but it still needs some work. However, I have some comments which are as follows:
• It will be helpful for the reader if in the introduction section the authors incorporate what is already known in the literature about the correlation between APOE4 and ABCA1 in ICC progression (such as in reference 21, revised version).
• Line 124-126: In this statement, I would like the authors to specify the lipid metabolic pathways they want to point out for the therapeutic strategy such as mentioned in reference 31(revised version). I am also curious to know if any ABCA1 drug is known in the literature for any cancer. Because that will make the impact of this work higher.
• Line 227-229: Please catalog the antibodies and their dilutions used for the experiments, that will be helpful for the reproducibility of the experiments.
• Line 325: It would be helpful for the readers if the authors elaborated the Figure 4D-4I in detail.
• Line 342-344: Same as the previous point, please elaborate on Figure 5I-5M in detail.

Experimental design

N/A

Validity of the findings

N/A

Additional comments

N/A

---

## Round 0.3 · accepted · Accept

All remaining issues were adequately addressed and the manuscript was amended accordingly. Therefore, the revised version is acceptable now.